# A Method to Present and Analyze Ensembles of Information Sources

**DOI:** 10.3390/e22050580

**Published:** 2020-05-21

**Authors:** Nicholas M. Timme, David Linsenbardt, Christopher C. Lapish

**Affiliations:** 1Department of Psychology, Indiana University—Purdue University Indianapolis, Indianapolis, IN 46202, USA; clapish@iupui.edu; 2Department of Neurosciences, University of New Mexico School of Medicine, Albuquerque, NM 87131, USA; dlinsen1@gmail.com; 3Stark Neuroscience Research Institute, Indiana University—Purdue University Indianapolis, Indianapolis, IN 46202, USA

**Keywords:** information theory, information ensemble, ensemble comparison, population coding, mutual information, neural ensemble, genetic network, population study

## Abstract

Information theory is a powerful tool for analyzing complex systems. In many areas of neuroscience, it is now possible to gather data from large ensembles of neural variables (e.g., data from many neurons, genes, or voxels). The individual variables can be analyzed with information theory to provide estimates of information shared between variables (forming a network between variables), or between neural variables and other variables (e.g., behavior or sensory stimuli). However, it can be difficult to (1) evaluate if the ensemble is significantly different from what would be expected in a purely noisy system and (2) determine if two ensembles are different. Herein, we introduce relatively simple methods to address these problems by analyzing ensembles of information sources. We demonstrate how an ensemble built of mutual information connections can be compared to null surrogate data to determine if the ensemble is significantly different from noise. Next, we show how two ensembles can be compared using a randomization process to determine if the sources in one contain more information than the other. All code necessary to carry out these analyses and demonstrations are provided.

## 1. Introduction

Information theory [1] is a valuable tool for analyzing complex systems. Information theory can quantify interactions in nonlinear systems, it can be applied to different types of data (e.g., continuous, discrete, and population signals [2]), it can be used to analyze multivariate systems, and it produces results in general units of bits that can be compared across systems.

In addition to analyzing the behavior of individual neural variables (e.g., a single neuron, voxel, or gene), evaluating the behavior of ensembles of variables (e.g., many neurons, voxels, or genes), is an increasingly tractable goal. These analyses quantify pairwise functional connections in networks by determining the amount of information shared between each variable. Importantly, these pairwise information measures can form the basis of subsequent analyses that seek to describe structural or functional connectivity within a network (e.g., [3,4,5,6,7,8,9,10,11,12], see [13] for discussions of population analyses in neuroscience). In addition, mutual information can be calculated between neural variables and other variables, such as environmental stimuli or behaviors, to provide an analysis of neural encoding by the ensemble. For the purposes of this paper, we will refer to both of these as ensembles of information sources (e.g., mutual information connections between neurons or mutual information values between neurons and a sensory stimuli). Furthermore, though we will utilize mutual information for demonstration purposes herein, the methods we will describe generalize to any ensemble of information sources, such as entropy, mutual information, or transfer entropy values. See [14] for an excellent recent treatment of the more specific question of inferring connections between neural variables using transfer entropy.

In this article, we will focus on two questions that arise when analyzing ensembles and how they can be addressed by considering the ensemble of information sources. First, is there a real ensemble in the data or just a noisy system? Using information sources, we can address this question by asking if the sources in the ensemble actually provide non-null information results. Second, are two ensembles different? By comparing the distribution of information sources in the two ensembles, we can evaluate if the information sources are significantly different. Importantly, the analyses we will discuss center on these questions, not the conceptually distinct and important question of how networks differ in structure, which is traditionally assessed with various topological measures like clustering coefficient and modularity [15].

When analyzing real data, understanding how basic features of the data (e.g., ensemble size, number of observations, discrete vs. continuous data, etc.) affect the assessment of statistical significance can be difficult. Indeed, how the basic features of data bias information theory analyses has been widely discussed in the literature [14,16,17,18]. While some methods exist to address these issues in certain cases (e.g., [19]), general solutions to this problem typically involve generating null surrogate data to estimate the probability that a given information value from a single information source was the result of noise and bias (i.e., a *p*-value) [20] (e.g., [8,21]).

Using a null surrogate data testing approach works well in the two following cases. First, when only a few sources are being analyzed, it is feasible to perform direct comparisons between those few sources and/or their associated *p*-values. Second, when only very strong signals are of interest, it is possible to set a threshold and limit the analysis to sources that yield sub-threshold *p*-values. However, in other cases, these methods present problems. As the number of sources increases, calculating *p*-values using null surrogate data becomes computationally expensive due to the number of sources and the improved resolution necessary to utilize multiple comparison correction techniques. Furthermore, it may be the case that the information sources do not possess extremely low *p*-values, but still weakly encode some information in some non-null fashion. These situations motivate the need for the novel statistical approaches proposed herein.

We introduce a general method to present, analyze, and compare data from many information sources. This is demonstrated with a relatively simple method to determine if an ensemble of information sources is distinguishable from a null distribution of information values (e.g., the ensemble contains more information than we would expect by chance), as well as a method to compare information source ensembles to determine if the distributions of information values are different from one another (e.g., one ensemble contains more information than the other). To demonstrate these methods, we will use an ensemble of mutual information values generated from a simple system. Appendix A necessary to carry out all of the demonstrations performed in this article is included as Appendix A and is posted on GitHub [22,23]. In addition to the demonstrations of these methods contained in this article, this method has also recently been used to assess neural encoding by large groups of neurons in the medial prefrontal cortex in rats [24].

## 2. Materials and Methods

### 2.1. Individual Information Source

In order to examine ensembles of information sources, we must first describe an individual information source. Each source is itself a system that could produce a wide range of information results (e.g., entropy, mutual information, or transfer entropy values). We have chosen to use a simple system and mutual information values associated with this system as the individual information sources in this study. We wish to emphasize that this simple system is only being used to generate data to demonstrate methods for analyzing ensembles of information sources.

The model system consists of two discrete variables (X and Y), each with only two states (0 and 1 where individual states of X and Y are noted as x and y, respectively). We could imagine that the X and Y variables represent the spiking state (spike vs. no spike) of two neurons. Thus, x=1 would correspond to neuron X spiking and y=0 would correspond to neuron Y not spiking. The mutual information between the neurons could then represent the strength of the connection between the neurons in a network. Importantly, the X and Y variables could be other pairs of neural signals (e.g., blood-oxygen-level-dependent (BOLD) signal values, electroencephalography (EEG) voltage), a neural signal with a non-neural signal (e.g., neuron spiking and a visual stimuli), or two other signals (e.g., gene expression for a pair of genes) because the information theory-based approaches presented herein are highly generalizable across systems. This point can be somewhat confusing when considering interactions between neural variables. In this case, the information sources are the interactions between the neural variables. In other words, the “ensembles of information sources” we refer to here are the connections between the neural variables in a network.

Because our primary interest is developing methods for use with experimental data and experimental data usually consist of multiple observations (i.e., trials) of a system, we assume data from the model contains $n_{obs}$ number of joint observations of both variables. To simplify the mathematics of the model, we further assume that $n_{obs}$ is a multiple of 4, but this is not a critical assumption for the analysis. Furthermore, for both observations of X and Y individually, we assume that half the observations produced a state of 0 and the other half produced a state of 1. By controlling how the states of X and Y are jointly related, we can control the strength of the interaction between X and Y, which will allow us to control the strength of the mutual information observed between X and Y. In addition, by controlling the number of observations, we can explore how this critical experimental parameter influences the detection of significant results.

Specifically, the strength of interaction between X and Y is controlled by the interaction strength variable s, which can range from 0 to 1. The number of joint observations of the states of X and Y is shown in Table 1. In this simple model, when s=0 (i.e., no interaction is present between X and Y), each joint state of X and Y is equally likely (e.g., the distribution across each cell is uniform). When s=1 (i.e., there is the strongest possible interaction between X and Y), half of the observations consist of the joint state (x=0,y=0) and the other half consist of the state (x=1,y=1), which implies that the state of one variable completely determines the state of the second variable.

To simulate experimental noise in the system, the joint state observations can be randomized using a noise variable a, which can range from 0 to 1. The randomization processes proceed by randomly selecting (uniform likelihood) $Round\left(n_{obs}a\right)$ joint observations and randomly permuting the Y variable state among the selected joint states. Therefore, when a=0, there is no noise in the system and it is only governed by the interaction strength variable s. When a=1, the system is completely dominated by noise and the interaction strength variable s has no impact on the eventual joint observations.

To produce a mutual information value for this simple system, we estimate the joint probability distribution $p_{dist}\left(x,y\right)$ by dividing the distribution of joint observation counts by the total number of observations ($n_{obs}$) (Note, we refer to probability distributions as $p_{dist}$ to avoid confusion with *p*-values from statistical tests, which we simply note as p). We then calculate the mutual information using Equation (1):(1)$$I\left(X,Y\right)=\underset{x\in X,y\in Y}{\sum }p_{dist}\left(x,y\right)log_{2}\left(\frac{p_{dist}\left(x,y\right)}{p_{dist}\left(x\right)p_{dist}\left(y\right)}\right).$$

When assessing information theory values, significance testing is critical. This is due to the fact that the model data (and real experimental data) contain a limited number of observations and mutual information must be greater than or equal to zero [1]. This leads to a situation where even models with no interactions will produce non-zero mutual information results due to noise and/or the finite number of observations. To account for this effect, we assessed the statistical significance of a mutual information result for each model using Monte Carlo techniques by comparing the mutual information produced by the original model to the distribution of mutual information values produced by many randomized null surrogate model data sets. The distribution of mutual information values from randomized data estimated the likelihood to observe a given mutual information value by chance even with no interactions present given the number of observations and the marginal distributions.

The randomization was accomplished by randomly permuting the Y variable state for all joint observations. This process preserved the number of observations ($n_{obs}$) and the underlying marginal distributions. (In effect, the null data consisted of models with matching $n_{obs}$ and a=1.) The number of null surrogate data sets ($n_{MC}$) was typically set to 100, 500, or 1000 (see below). The fraction of null surrogate data sets that produced mutual information values larger than or equal to the mutual information value from the original model was used as an estimate of the *p*-value for the mutual information result. For example, if the original model produced a mutual information value of 0.2 bits and 30 out of 1000 null surrogate data sets produced mutual information values equal to or larger than 0.2 bits, the *p*-value was estimated as p=0.03. If no null surrogate data sets produced mutual information values larger than or equal to the original model’s mutual information result, we estimated the *p*-value as $p=1/\left(2n_{MC}\right)$.

### 2.2. Information Source Ensemble Analysis

So far, we have described a simple individual model system to examine, methods to analyze an information theory measure from this system (in our case, mutual information), and methods to assess the statistical significance of that information theory result. Now, we will describe general methods to assess the behavior of many information values produced by ensembles with $n_{net}$ individual information sources. These methods do not rely on the specifics of the individual model system. Rather, these methods only require that the ensemble consist of individual elements, each of which possesses an information value and a collection of null information values derived from that source, from which a *p*-value can be calculated. In general, assume we have a system of $n_{ens}$ individual information sources where each individual information value is noted as $I_{i}$ where $i=1,2,3,\ldots n_{ens}$. Furthermore, associated with each information value is $I_{null,i,j}$ null information values (where $j=1,2,3,\ldots n_{MC}$) and a *p*-value $p_{i}$. The relevant parameters for our model are shown in Table 2. To demonstrate these methods, we generated model data for three ensembles with varying levels of noise (Figure 1).

The first question we wished to ask about the ensemble of information sources is whether the ensemble itself produced significant information results. Indeed, even with an ensemble of sources dominated by noise, some sources will randomly produce low *p*-values (i.e., false positives). Obviously, this is an important experimental question. For instance, one might wish to know if a group of neurons significantly encodes some sensory stimuli or whether the group of neurons significantly share information (i.e., form a network).

To determine if the ensemble produces significant information results, we performed a Kolmogorov–Smirnov (KS) test between the information values from the real information sources and the null information values produced in the significance testing of the individual information sources (i.e., the set $\left\{I_{i}\right\}$ compared to the set $\left\{I_{null,i,j}\right\}$). In the three example ensembles shown in Figure 1, the low noise ensemble produced an information measure distribution very different from the null data (Figure 1A1,B1), which resulted in a low *p*-value from the KS test between the distribution of real information measures and the null data (Figure 1D). Conversely, an ensemble dominated by noise (Figure 1A3,B3) produced distributions of real and null information measures that were very similar and, as a result, a high *p*-value via KS test (Figure 1D).

Three important points should be made about this method. First, at the current time, ensembles that consist of individual systems governed by the same rules (i.e., homogenous systems) are required for this approach. In the future, we hope to fully characterize how this method handles ensembles of heterogeneous systems (see Section 4.3), but we wish to emphasize that homogenous systems are a requirement of this method at this time.

Second, the use of the KS test allows for the detection of information distributions that might be smaller than expected given the null distribution. For instance, if the information values are skewed to be smaller than expected by chance, or if the distribution is bimodal but has a mean value near the mean for the null data, this method will allow for the detection of information results significantly lower than null for an ensemble. In this way, this method is useful to detect the suppression of information in an ensemble.

Third, this method does not require generating large numbers of surrogate null information values because it does not seek to assess the significance of each information source. The null information value distribution will have $n_{ens}*n_{MC}$ values and the original data information distribution will have $n_{ens}$ values. Therefore, relatively few null information values per information source (perhaps as few as 10) should be sufficient to perform the KS test between the null information distribution and the distribution of the real information values.

Next, we sought a method to conveniently and compactly present the information values produced by an ensemble. The most complete presentation method would be to show the full distribution of observed information results, but doing so is not ideal in many circumstances. For instance, when examining the time evolution of an information ensemble, it would be confusing to present distributions for each time point and attempt to compare distributions through time. Furthermore, it may be additionally difficult to present distributions along with *p*-value information, though a scatter plot or 2D histogram would be options to present both values simultaneously. Therefore, we developed a method of presenting the weighted mean, weighted standard error of the mean, and weighted standard deviation.

In order to calculate these weighted quantities, it was first necessary to create weights. We chose to use the *p*-value for each individual information value (see Section 2.1) in the ensemble to calculate the weight for that information value via Equation (2) (Figure 1C):(2)$$w_{i}=-log_{10}\left(p_{i}\right).$$

The weights were normalized by dividing each weight by the sum of all weights. This method of weighting the data allowed for information values with lower *p*-values to exert more influence on the mean or error statements for the whole ensemble. We chose to perform this weighting to bias the mean and error statements in a manner that more closely reflects the information sources that are more likely to be significant (i.e., have a lower *p*-value). Furthermore, this weighting method does not require large numbers of null surrogate data to be generated. Given the methods used to determine whether an ensemble of information sources is significant (see above) and if two ensembles are significantly different (see below), we did not assess family-wise error in these *p*-values or these weights.

Next, the weighted mean (Equation (3)), weighted standard error of the mean (Equation (4)), and weighted standard deviation (Equation (5)) can be calculated and presented to convey general features of the ensemble of information sources (Figure 1D):(3)$$\bar{I_{i}}=\overset{n_{ens}}{\underset{i}{\sum }}w_{i}I_{i},$$
(4)$$\sigma_{\bar{I_{i}}}=\sqrt{\frac{1}{n_{net}-1}\overset{n_{ens}}{\underset{i}{\sum }}{\left(I_{i}-\overset{n_{ens}}{\underset{i\;}{\sum }}I_{i\;}\right)}^{2}*\overset{n_{ens}}{\underset{i}{\sum }}w_{i}^{2}},$$
(5)$$\sigma_{I_{i}}=\sqrt{\overset{n_{ens}}{\underset{i}{\sum }}{\left(w_{i}\left(I_{i}-\overset{n_{ens}}{\underset{i\;}{\sum }}w_{i\;}I_{i\;}\right)\right)}^{2}}.$$

Note that this weighting method will not correct for family-wise error and will still possess bias in information values resulting from randomly observed high information values (i.e., false-positives).

The third question we wished to ask is how we can assess whether two ensembles of information sources are significantly different. This question might arise experimentally when a researcher wishes to examine the effects of a treatment. For instance, one might want to quantify if a group of neurons in one treated subject share more or less information than an ensemble from a control subject. Towards this goal, a similar Monte Carlo approach as outlined previously was utilized. We generated null surrogate data of differences in the weighted mean between ensembles of information sources by randomly permuting individual information values and their associated weights between ensembles while preserving the number of information sources in each ensemble. We estimated the *p*-value as the proportion of null comparisons with weighted mean differences greater than that observed in the comparison between the real ensembles while accounting for the sign of the difference (i.e., ensemble A greater than ensemble B or vice versa). In the example systems shown in Figure 1, this ensemble comparison method produced a *p*-value of p=0.038 when comparing the two lower noise ensembles that had more similar weighted means, but lower *p*-values ($p<10^{-3}$) for comparisons with the highest noise system.

### 2.3. Software

MATLAB software is included with this paper and available freely on GitHub [22,23] to produce the model data discussed in this article as well as the figures in the manuscript. Given the simplicity of the model, the software is not complicated. Data from the model is generated using the function netModel.m. Each figure is produced by a stand-alone script named after the figure. For instance, the script Figure1.m will produce Figure 1.

## 3. Results

In order to evaluate the performance of the methods described herein, we explored significance results across a wide range of parameters. However, we wish to emphasize that these specific parameters are most relevant to the simple model we used to demonstrate these methods. In other systems, with different numbers of states or different types of dynamics, these results may not hold.

First, we were interested in evaluating how the signal strength, noise, the number of sources in the ensemble, and the number of observations interacted to produce different weighted means and errors in each ensemble (Figure 2). This analysis produced results that largely agree with expectation: increased noise produced ensembles that were not significantly different from null, as did decreased signal strength.

Importantly, these results can provide intuition about experimental parameters that can be varied like number of observations and information sources (though recall that different underlying dynamics may affect results). This could be useful when determining power in an experimental design and inform decisions on sample size. Overall, we find that even for few observations and information sources, results significantly different from null can be detected for low noise and/or high signal strength systems. As one would expect, adding more sources to the ensemble and adding more observations improves the ability of the analysis to detect significant information results. However, these results demonstrate that it is not necessary to record from an ensemble of 160 information sources over 100 trials to detect significant differences. For instance, even with 40 information sources and 20 trials, significant results were found for a wide range of signal strengths, even in the presence of noise. It should be noted that for ensembles with large amounts of noise with few sources and observations, a bias in the weighted mean mutual information was observed due to randomly observed high information sources (i.e., false-positives), though this bias did not result in significant ensembles being detected.

Next, we further explored the ability of this analysis method to detect ensembles of information sources with significantly low amounts of information. When we examined one example set of parameter values, we found that some low values of signal strength and noise produced information results that were significantly smaller than the null (Figure 3). For this example, number of ensemble sources and observations, we found that signal strengths between 0.08 and 0.2 produced ensembles with information values not significantly different from null for a=0.2 and a=0.3. However, for lower and higher signal strength values, the ensemble produced information values significantly different from null. In the case of large signal strengths, this result can be understood simply as signal overwhelming noise to produce significantly larger information results in the ensemble. In the case of small signal strengths, the noise in the system is weaker than in the null model and the ensemble of information values is skewed heavily towards small information values. Thus, the distribution of information sources from the ensemble is more skewed towards low information values in comparison to the null (fully randomized) data, producing a significantly lower information result.

Finally, we examined how parameter values affect comparisons between ensembles. For example, one may be interested in detecting if the effect of some treatment on subjects is different than some control. In the context of ensembles of information sources, such a comparison can be performed by asking whether two ensembles are significantly different. If the information sources consist of connections between variables in the ensemble, then the comparison is between the distributions of connection strengths, which can be helpful when ensemble node identity cannot be established between ensembles (e.g., neurons recorded from different animals). If the information sources consist of encoding performed by the neural variables, then the comparison is between distributions of neural encoding strengths. At a fixed level of moderate noise (a=0.5), we examined how the size of the ensembles and the number of observations affected the ability to detect significant differences between ensembles with different signal strengths (Figure 4). (As expected, for small ensembles and few observations, even comparisons between ensembles with very different signal strengths produced relatively high *p*-values. However, for larger ensembles and more observations, comparisons between ensembles with different signal strengths yielded low *p*-values.

## 4. Discussion

The primary results of this article are the development of techniques (1) to compactly communicate important features of the distribution of information values associated with an ensemble of information sources, (2) to determine if a distribution of information values is significantly different from null, and (3) to determine if two ensembles of information sources are significantly different. Novel methods to assess large complex data are required as methods to create ‘Big Data’ are increasingly more commonplace. With the proliferation of these types of data, formalizing statistical approaches for their characterization is critical for rigor and reproducibility. Critically, the methods outlined here provide both descriptive and inferential statistical approaches.

### 4.1. Method Generalizability

We used a very simple system of binary variables to generate data to which the information source ensemble analyses could be applied. Of course, many systems of interest possess more than two discrete states or are continuous. For instance, a researcher may be interested in the number of action potentials a neuron produces in a given time bin and so discretize by the number of action potentials. In addition, a researcher may be interested in a continuous variable, such as BOLD signal from fMRI, and discretize the data into some number of bins or use some other method for analyzing continuous data. Furthermore, many other systems possess more complex relationships between variables, as well as more than two variables. For instance, transfer entropy analyses utilize three variables and are often applied to systems with non-linear relationships. However, the ensemble analysis methods outlined in this article could be applied to any type of information values (e.g., entropy and mutual information [1,25], transfer entropy [26], or synergy and redundancy [20,27,28]) generated by any type of system, so long as each information value is associated with a set of null information values from which a *p*-value can be calculated. In general, it is possible to follow the methods described in Section 2.2 for a system with different dynamics and/or a different information measure, so long as each element of the ensemble possess an information value and a set of null information values from which a *p*-value can be calculated. To be clear, the underlying dynamics of the systems (e.g., more than two states or a continuous system) and the type of information measure (e.g., transfer entropy instead of mutual information) will affect the end results and interpretation. Given the wide range of analysis options available, we felt it was beyond the scope of this article to demonstrate these other types of analyses herein, though we discuss some other possible uses below.

As an example, suppose a researcher wished to use the ensemble analysis methods outlined herein to study transfer entropy values between many time series (e.g., multiple neuron spike trains or multiple EEG electrodes). That researcher would first pick a method to evaluate the transfer entropy for each pair of time series. In addition, the researcher would need to create null data (via spike train jittering or some other method) that could be analyzed using the same techniques to obtain null transfer entropy values for each pair. This initial stage of the analysis would require many decisions based on the type of data and the effects the researcher wished to examine. Discussing this stage of the analysis for every type of data and every information measure is beyond the scope of this paper. However, once that information theory analysis is complete, the researcher could use the methods described in Section 2.2 to evaluate the significance of the transfer entropy values and compare ensembles of transfer entropy values. Specifically, the transfer entropy values would be $I_{i}$, the null transfer entropy values would be $I_{null,i,j}$, and the associated *p*-value would be $p_{i}$.

Of course, choices made in the initial analysis would affect the outcome of the ensemble analysis. For instance, if the researcher used time bin sizes that were too small and no significant transfer entropy values were found, the ensemble analysis will show that the ensemble was not significantly different from null. Or, if the researcher used an inappropriate null model, the ensemble analysis may incorrectly find an ensemble that is significantly different from null. Therefore, the results of an ensemble information source analysis using the methods described in this article must always be interpreted using knowledge of the underlying information theory analysis of the individual information sources.

### 4.2. Relationship to Neural Systems and Neural Networks

It is important to emphasize that these techniques could be used with ensembles of information sources generated by other types of systems (e.g., groups of genes, patients, consumers, communication channels, etc.). That said, we feel that these methods for analyzing ensembles of information sources fill an important role in the analyses of large neural recordings in particular. State-of-the-art neural recording systems are currently capable of producing data for hundreds to thousands of neural variables. These recording techniques present an ever-growing challenge, especially given the distributed nature of computation in the neural systems where it may be necessary to examine large portions of a neural system simultaneously to understand how it functions.

In particular, these methods allow for the study of important features of networks. By considering the connections within a network to be the ensemble of information sources, it is possible to use these methods to ask whether a network exists. In this case, the ensemble would contain one information source for every connection in the network. If the information observed within a network is sufficiently different from that which would be expected by chance among disconnected nodes, then it is possible to conclude that the system under study is not simply disconnected nodes acting independently, but rather something larger: a network. This is an important first hurdle to any network analysis. Furthermore, using methods for comparing information sources, it is possible to compare the distribution of connections in two networks and thus provide a method for determining when networks are different that is independent of structure (e.g., clustering coefficient [15]) and node identity. Other analyses comparing structural measures (with or without identify of nodes) between networks are valuable tools that provide important information about how the structure of networks compare. In contrast, the methods described herein are able to compare the amount of information shared within networks using distributions of information values.

In addition to studies of networks of neural variables, several other common types of neuroscience experiments could utilize these analysis techniques as well. For instance, in an in vivo electrophysiology experiment that records spiking activity from many neurons simultaneously, a researcher might ask how information encoded by the neurons about a stimulus or a behavior changes through time during a task. At each time point in the task, the researcher could plot an average information encoded by the neurons along with a description of the spread of the information results (e.g., error bars). The weighted mean and standard error of the weighted mean technique utilized herein would accomplish that goal. Furthermore, the researcher would want to know if the ensemble of neurons are actually encoding information about the stimulus or behavior. This can be done by comparing the distribution of the information values from the ensemble to a null distribution of the information values expected by chance for a system with similar underlying statistics. Finally, the researcher might want to know if one population of neurons encodes more information than another population. These two populations could come from different subjects (e.g., treatment vs. control) or from different brain regions (e.g., prefrontal cortex vs. sensory cortex). Such a comparison can be performed using the randomization and weighted mean comparison technique outline herein. The techniques outlined in this article could address similar questions if the data arose from calcium imaging, EEG, or fMRI.

These methods were recently used to examine neural encoding in medial prefrontal cortex in rats [24]. In this study, the encoding of sensory stimuli and future behaviors by individual neurons was examined throughout a task to determine when significant encoding was occurring (non-noise ensemble of information sources) and encoding between two treatment groups was also compared (assessing differences between ensembles of information sources).

### 4.3. Limitations and Future Research

While the methods described in this article are the first general techniques for analyzing, presenting, and comparing ensembles of information sources we are aware of, there are several limitations to these methods and possible improvements that can be pursued in the future. First, the ensemble analyses presented herein do not consider additional information about the information sources beyond the information value, the *p*-value, and the null surrogate data. In the case of analyses of networks, the identity of the connection could be important to the study. For instance, certain connections (e.g., pairs of neurons involved in projections from one brain region to another) could be of special interest and necessitate dividing the ensembles under study into sub-ensembles for analysis with these methods. In the future, additional approaches could be developed that integrate more sophisticated methods for including additional data about information sources in these methods.

Second, these methods require generating large numbers of null surrogate data to assess *p*-values, both for individual sources and between ensembles. Generating these null data can be computationally expensive. It would be advantageous to develop techniques to calculate *p*-values for various information measures analytically. If these advances were made, it would speed up finding the *p*-value for individual information sources, but it would complicate the assessment of the ensemble’s significance because there would be no null data for comparison. Furthermore, this advance would not shorten the computation time to assess significant differences between ensembles.

Along these lines, several researchers have proposed methods specifically designed to infer interactions among pairs of variables (e.g., ‘network inference’). As mentioned above, a specific recent example used transfer entropy to assess connections between neural variables [14]. These methods frequently seek alternative methods to generating large numbers of null surrogate data for each possible connection among variables (see, for instance, [29,30,31]). In contrast, the methods presented herein require assessing every possible connection within the network. We sought to develop slightly more general methods that could be used on ensembles of information values generated in different ways, such as transfer entropy values between pairs of neural variables, but also mutual information values between neural variables and sensory stimuli. Furthermore, we sought to create a method capable of reducing the number of null surrogate data that had to be generated. Still, the more specialized network inference methods referenced above may be more efficient than the methods we propose for those specific types of analyses, so the interested reader is strongly encouraged to explore those alternative methods.

As discussed in Section 2.2, we have not examined the behavior of these methods for heterogeneous systems. We believe these methods do not require homogeneous systems for the following reason. Since the null data are derived from the parameters of each individual system, the significance assigned to each information value will be determined by the parameters of the individual system. This is because the null distribution for the ensemble is populated with null data that maintains the relevant statistics for each individual source in the ensemble (e.g., number of observations, marginal distributions, etc.). For instance, in terms of the simple demonstration model used here, in the future, we will assess if it would be possible to use this method on a system with heterogeneous values of s. This feature would make this method suitable for heterogeneous systems such as mixtures of different cell types or voxels that are governed by different hemodynamic response functions. In the future, we hope to examine precisely how heterogeneous systems affect the outcome of these types of analyses. These studies will assess several points, such as how well can an ensemble with just a few strong sources be detected and can bias in some sources blind the method to detecting an ensemble composed of weaker sources.

Finally, these methods have been developed for networks of information sources characterized by single values (e.g., a single mutual information value for each source). Future research could be conducted to expand these methods to multivariate information sources (e.g., multiple partial information values [27] (synergy, redundancy, etc.) for each information source within the ensemble). In the future, we hope to demonstrate these methods for more information measures, numbers of bins, and methods for handling continuous data.

## Figures and Tables

**Figure 1 entropy-22-00580-f001:**
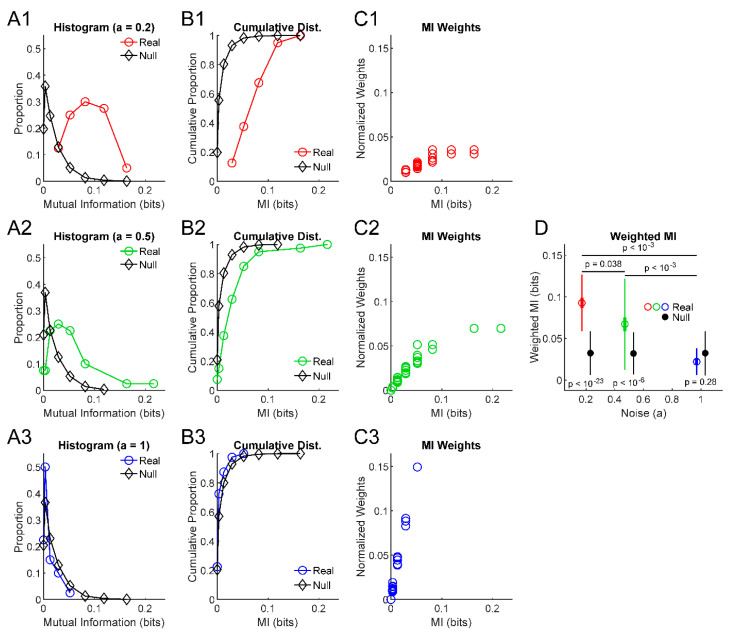
Assessing ensemble difference from null and differences between ensembles. (**A**) Histograms of mutual information results from example ensembles of information sources and the null information results from these sources produced through randomization ($n_{ens}=40$, $n_{obs}=60$, s=0.4, $n_{MC}=100$). (**B**) Cumulative distributions of the information values from (**A**). (**C**) Scatter plots of normalized weight values (-log_10_(p)) for each information result. (**D**) Weighted mean (dot), weighted standard deviation (thin lines), and standard error of the weighted mean (thick lines) for the example ensembles, along with the same results for the null data produced by randomization. Note that the lower noise examples (1 and 2) produced very low *p*-values in comparison to null data via KS tests (*p*-values shown below error bars, see (B) for reference). Still, note that null data produced non-zero weighted mean mutual information values. Comparisons between the weighted means of the example ensembles via randomization produced low *p*-values between all pairs (*p*-values shown above error bars), though resolution was limited by the number of Monte Carlo trials (1000) to $p<10^{-3}$.

**Figure 2 entropy-22-00580-f002:**
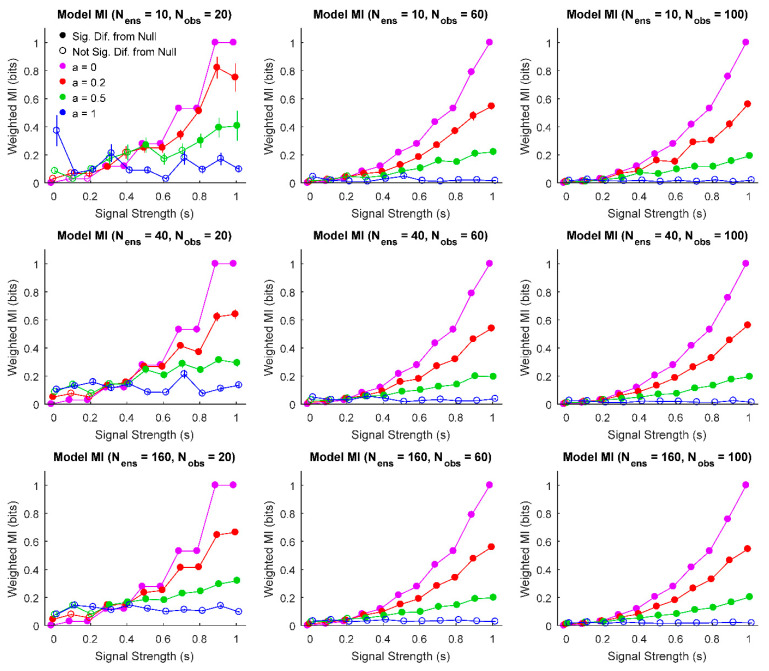
Weighted mean, standard error of the weighted mean, and comparison to null behavior for various model parameters. For a wide range of parameter values, information values increase with increased signal strength (as expected). Furthermore, high noise and/or low signal result in a lack of significant difference from null (significance threshold: p<0.01). Larger ensemble sizes and numbers of observations also decrease the uncertainty in the weighted mean as measured by standard error of the weighted mean. Due to the discrete nature of the model, increases in the number of observations results in smoother information curves because more observations allow for more possible model probability distributions for individual information sources. (Signal strength values jittered slightly to improve legibility).

**Figure 3 entropy-22-00580-f003:**
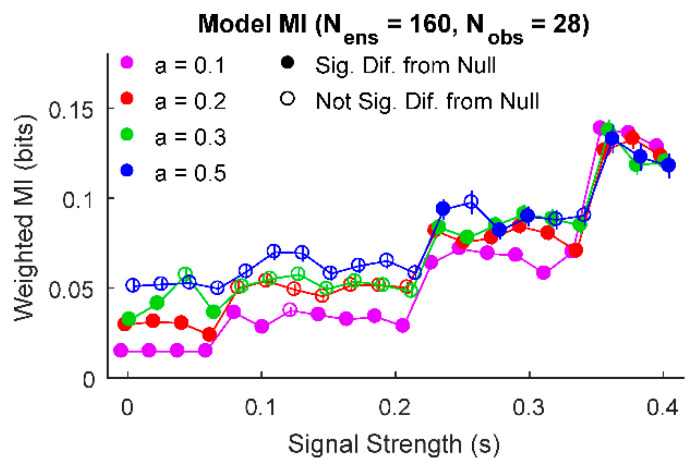
Information source ensembles can have significantly low amounts of information. For some parameters (e.g., a=0.2 and a=0.3), information source ensembles exhibited significant difference from null for low and high signal strengths (e.g., s<0.08 and s>0.2), but not over a middle range of signal strengths (e.g., 0.08≤s<0.2). (Signal strength values jittered slightly to improve legibility).

**Figure 4 entropy-22-00580-f004:**
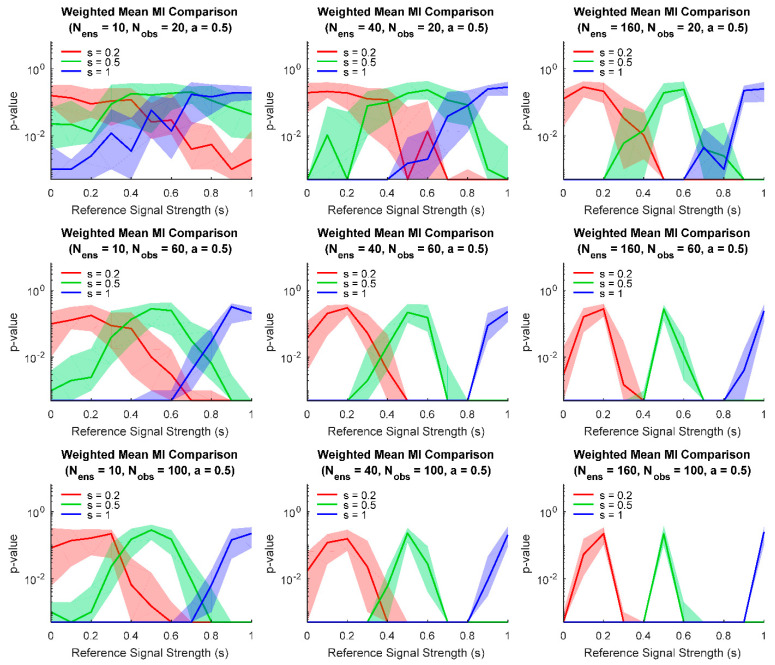
Comparisons between ensembles for various model parameters. Numerous models (signal strengths given on the horizontal axis) were compared to other models with three possible signal strengths (red, green, blue). As expected, for small ensembles and/or few trials (upper left), ensembles with similar signal strengths produce high *p*-values when their weighted mean values were compared using randomization. When larger ensembles and/or more trials are used (lower right), comparisons are able to detect smaller differences in signal strength. (Fifty model pairs generated for each parameter pair, line: median of these pairs, fringe: interquartile range of these pairs. One thousand randomization trials were performed to calculate *p*-values, resulting in a minimum *p*-value of 0.0005.).

**Table 1 entropy-22-00580-t001:** The number of joint observations for a single information source system based on the total number of observations and the interaction strength s between the variables X and Y. The Round(x) operation rounds the argument to the nearest integer (numbers with fractional elements equal to 0.5 are rounded to the next largest integer).

	x=0	x=1
***y* = 0**	$\frac{n_{obs}}{4}+Round\left(\frac{n_{obs}}{4}s\right)$	$\frac{n_{obs}}{4}-Round\left(\frac{n_{obs}}{4}s\right)$
***y* = 1**	$\frac{n_{obs}}{4}-Round\left(\frac{n_{obs}}{4}s\right)$	$\frac{n_{obs}}{4}+Round\left(\frac{n_{obs}}{4}s\right)$

**Table 2 entropy-22-00580-t002:** The meaning of the relevant model parameters.

	Meaning
***s***	The strength of the interaction (0: no interaction, 1: strongest possible interaction)
***a***	The noise level (0: no noise, 1: only noise)
$n_{ens}$	Number of information sources in the ensemble
$n_{MC}$	Number of randomization (Monte Carlo) trials in the null data comparison

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
