# Peer review of "A Method to Present and Analyze Ensembles of Information Sources"

_entropy, 2020, doi:10.3390/e22050580_

Round 1

Reviewer 1 Report

I thank the authors for their clarification of the issues raised in the first review. In my view, the paper greatly benefits from the change in terminology (network to ensemble), which makes the difference between the proposed method and existing network inference literature much clearer. It also resolves most of the major issues raised in the first review. A further main point regarding the applicability to heterogeneous systems has been thoroughly addressed by the authors (a comment on which heterogeneous features influence estimates, e.g., sample size or variable dimension, could be added in the discussion).

Thank you also for checking with the editor to fix references within the Word document. This increased readability significantly.

I have no further comments on the revised version of the paper and suggest the acceptance of the manuscript. In my opinion, the presented method addresses a gap in the information-theoretic analysis of multi-variate systems and is thus an interesting and important contribution to the field.

Reviewer 2 Report

I have no further comments

This manuscript is a resubmission of an earlier submission. The following is a list of the peer review reports and author responses from that submission.

Round 1

Reviewer 1 Report

The authors Timme, Linsenbardt, and Lapish present a method to establish statistical significance of single networks and differences between multiple networks; they furthermore introduce measures to compactly describe features of networks inferred with information-theoretic measures. The topic is interesting and highly relevant for disciplines concerned with the inference of networks from multivariate data using information-theoretic methods. Statistical significance testing is here mandatory due to the known and well illustrated problems when estimating information theoretic measures, e.g., mutual information, from noisy and/or limited data.

The manuscript is well written and structured, and in general easy to follow. However, I propose some major revisions and additional experiments bevor the paper is suitable for publication. Major points are listed in the following, below I list minor corrections and typos.

1. The presented method is potentially of high interest to a lot of disciplines employing network inference and analysis, and it is especially promising for data sets with large numbers of variables (as discussed in section 4.2). However, especially in neuroscience, a lot of research on similar topics exist. The authors should make the differences and advantages of their approach compared to other methods clearer. For example, a recent paper by Novelli et al. (2019) introduces a method for controlling the family-wise error rate in network inference. The present manuscript aims at a similar goal and should therefore compare the presented method to the approach by Novelli et al. Another example using

2. Along similar lines, the authors should evaluate and report the scaling behavior of their algorithm. For example, when calculating the mutual information for each of the n_net nodes, n_net-1 mutual information values would have to be calculated per node. This amounts to O(n_MC * n_net^2) MI-estimations per network. Is this understanding correct? If so, this number scales rather unfavorably and in general, network inference algorithms try to avoid estimating and testing the full number of possible connections (e.g., Runge (2012, Phys Rev Lett (108)25); MuTE (2014, PLoS ONE 9(10)); Wollstadt (2019, JOSS, (4)34)). An evaluation of the theoretical and practical run times as a function of network size would be desirable for the presented approach.

3. In section 2.2, the authors discuss that the proposed method for determining if a global "network effect" exists is especially suitable for heterogeneous networks. I am wondering, if this statement is true in general. For example, some properties of the data influence the bias of an information-theoretic estimate, such as the number of samples or dimensionality of variables when using the popular estimator by Kraskov et al. (2004, Phys Rev E, (69)6) to estimate mutual information. Here, if two nodes in the network have unequal numbers of samples or dimensionalities, the bias in estimates will differ widely. Accordingly, the bias in null information values will vary. I am wondering if this doesn't affect the proposed approach. One could think of a simple example, where one node produces high estimated information values due to high bias, which do not necessarily coincide with true mutual information (e.g., because the number of samples is lower compared to other nodes). For this node, we will obtain n'_MC null values that are likely larger in magnitude than the individual information values of all other nodes in the network. Won't this single node bias the whole procedure because now n'_MC null values are larger than n_net-1 information values in the network, simply due to a different bias?

The bias also depends on the strength of relationships (see Kraskov, 2004, Fig. 2), where the bias seems to be highest for strong relationships. So even if the number of samples and dimensionality of variables were held constant, information values may vary in bias due to different interaction strengths. Furthermore, the dependency of the bias not only holds for the estimator by Kraskov but also for example for plug-in estimators as for example the authors show in a previous paper (Timme, 2008, eNeuro, (5)3).

4. In section 2.2, the authors perform experiments to demonstrate the application of their method. Here it would be helpful to get more details on how the information measures used were obtained (e.g., which estimator used). Furthermore, the mutual information is used as the information measure of interest. Here, to me it was not clear how this resulted in a single measure per node exactly. In my understanding, the mutual information would have to be calculated between every pair of nodes in the network, resulting in n_net(n_net - 1) measures in total and (n_net - 1) information values per node, instead of a single one.

# Introduction:

The introduction should compare their method to approaches that explicitly try to handle network inference while controlling for the family-wise error rate (e.g., Novelli, 2019, Network Neurosci).

"estimate the likelihood that a given information value from a single information source was the result of noise and bias (i.e., a p-value)" -> replace "likelihood" by "probability"

TeX-References to tables and figures are not working.

When introducing s, write that s=1 means that the state of one variable completely determines the state of the second variable (while keeping the existing explanation). This is probably easier to understand.

# Materials & Methods:

"This leads to a situation where even models with no interactions will produce non-zero mutual information results due to noise and/or the discrete number of observations." -> replace discrete by finite?

Beginning of page 4: "When assessing information theory values, significance testing it critical." -> it to is

In section 2.2, eq. (2), the authors use the p-value of each node/pair of nodes to weight the distributions. Does this control for the family-wise error rate? The authors should report the true and false positives according to the p-values found for individual measures. Using uncorrected p-values may bias measures, wouldn't they? Based on these weights, general measures used to represent features of the network are introduced (eqs. 3-5). It would be helpful to have some illustration how these measures are to be interpreted, e.g., their range, what high/low values mean for the network in question, etc.

# Results:

When discussing results shown in Fig. 2, no discussion of potential influences of an artificially inflated false-positive rate is provided. Shouldn't this be considered when analyzing 160 nodes in parallel?

"If the information sources consist of connections between variables in the network, then the comparison is between the distributions of connection strengths, which can be helpful with network node identity cannot..." -> with to when?

# Discussion:

In 4.3, the authors write "the methods described in this article are the first general techniques for analyzing, presenting, and comparing networks of information sources". However, network comparison methods for information transfer and mutual information networks have been presented, for example, by Lindner et al. (2011, https://doi.org/10.1186/1471-2202-12-119) and in the successor toolbox by Wollstadt et al. (2019, https://doi.org/10.21105/joss.01081). Both toolboxes implement statistical tests for network comparisons and the comparison of individual links. The authors should compare their method against these approaches.

Reviewer 2 Report

The authors introduce an information-theoretic methodology for analysing the structure of networks of random variables (here referred to as "information sources"), such as, for example, neurones or genes. Specifically, they address the following two questions: a) is a particular network of variables different from a purely random system and b) are two particular networks different from each other? These questions could be of interest to researchers in various fields, including systems biology, computational neuroscience and genomics.  

The authors proceed by introducing networks of random binary variables, which are parametrised by network size, number of observations, levels of noise and interaction strength between pairs of component variables. They choose as measure of information the mutual information across any pair of variables, which they subsequently weight and aggregate across the whole network. 

The answers they propose to the two aforementioned questions rely on null hypothesis significance testing, where the null distribution is empirically constructed using a permutation methodology, an approach commonly used by applied statisticians. Overall, their results are not surprising: higher information content is detectable at lower noise levels, higher interaction strengths, larger networks and higher numbers of observations. Furthermore, the Monte Carlo methodology described here is relatively simple. Nevertheless, this is a well written and scientifically sound study and I hope the authors may find the points mentioned below useful:

1) It is surprising that the authors do not conduct any experiments using natural (i.e. not simulated) data. Their work is clearly motivated by problems in computational neuroscience and systems biology and the lack of actual experimental data in this paper is a glaring omission. In my view, it is necessary that the authors demonstrate how their work can be applied in the analysis of a small number of experimental datasets (possibly, in the public domain) of the type they refer to throughout their paper.

2) The authors focus on networks composed of binary random variables and they merely discuss how their approach can be applied to more general discrete variables or to continuous variables. In my view, this is not sufficient: they should develop specific models for the two aforementioned types of variables and, then, study them as they have already done for binary variables. Investigating different measures of information content other than mutual information is also desirable, but it could be the topic of a separate paper.                    

3) It is conceivable that networks are characterised by some degree of heterogeneity, which in this particular study may be encoded by treating the interaction strength s as a random variable, which varies across different pairs of information sources. Would introducing such heterogeneity modify the conclusions of the study and how?             

Minor points:

1) I could not run the code, since I do not have Matlab installed. I urge the authors to consider using open source software in the future, such as Python or R. Nevertheless, the code seems well organised and commented.

2) Line 102: "theory-bases" should be "theory-based"
